# Selectivity of Oxygen Evolution Reaction on Carbon Cloth-Supported δ-MnO_2_ Nanosheets in Electrolysis of Real Seawater

**DOI:** 10.3390/molecules28020854

**Published:** 2023-01-14

**Authors:** Haofeng Yan, Xuyun Wang, Vladimir Linkov, Shan Ji, Rongfang Wang

**Affiliations:** 1College of Chemical Engineering, Qingdao University of Science and Technology, Qingdao 266042, China; 2South African Institute for Advanced Materials Chemistry, University of the Western Cape, Cape Town 7535, South Africa; 3College of Biological, Chemical Science and Engineering, Jiaxing University, Jiaxing 314001, China

**Keywords:** MnO_2_, anode, oxygen evolution reaction, hydrogen production, seawater electrolysis

## Abstract

Electrolysis of seawater using solar and wind energy is a promising technology for hydrogen production which is not affected by the shortage of freshwater resources. However, the competition of chlorine evolution reactions and oxygen evolution reactions on the anode is a major obstacle in the upscaling of seawater electrolyzers for hydrogen production and energy storage, which require chlorine-inhibited oxygen evolution electrodes to become commercially viable. In this study, such an electrode was prepared by growing δ-MnO_2_ nanosheet arrays on the carbon cloth surface. The selectivity of the newly prepared anode towards the oxygen evolution reaction (OER) was 66.3% after 30 min of electrolyzer operation. The insertion of Fe, Co and Ni ions into MnO_2_ nanosheets resulted in an increased number of trivalent Mn atoms, which had a negative effect on the OER selectivity. Good tolerance of MnO_2_/CC electrodes to chlorine evolution in seawater electrolysis indicates its suitability for upscaling this important energy conversion and storage technology.

## 1. Introduction

Hydrogen, one of the most promising alternatives to fossil fuels, has recently attracted tremendous attention due to its high energy density, renewability and environmental friendliness [1,2,3]. Currently, hydrogen production using water electrolysis utilizes fresh water, which limits its applicability in arid regions. The electrochemical splitting of pure water requires substantial electrical input energy, since the resistance of pure water is 18 MΩ cm. In contrast, seawater is up to six orders of magnitude lower (R_Seawater_ = 20 Ω cm) which, viewed from the perspective of conductivity, in principle allows energy-efficient splitting of water [4]. Seawater resources account for about 96.5% of water on the planet and can be regarded as an infinite resource compared with fresh water [5]. The use of seawater instead of fresh water for electrolytic hydrogen production can play an important role in energy crisis alleviation, while conserving freshwater resources. Due to the complex composition of seawater, its electrolysis is still a major challenge for large-scale hydrogen production [6]. Specifically, seawater is rich in electrochemically active Cl^−^, which undergoes the chlorine evolution reaction (ClER) competing with the oxygen evolution reaction (OER) on the anode [7]. Concurrently, free chlorine and hypochlorous acid generated during Cl^−^ oxidation can corrode the electrode, resulting in poor performance of the electrolyzer [8]. ClER has a higher thermodynamic potential than that of OER, and their potential difference increases with rising pH [9], reaching 490 mV in an alkaline media [10]. Many electrocatalysts were developed to prevent hypochlorite formation during seawater electrolysis, such as S-doped Ni/Fe hydroxide [11], amorphous nickel-iron phosphide (NiFeP) hollow sphere CoFeSx [12] and Ag-doped NiFeLDH [13]. At present, concentrated KOH solution is used as an electrolyte for water electrolysis, which causes additional expense and environmental problems [14]. Using natural seawater as the electrolyte will greatly reduce the cost of large-scale hydrogen production [15]. Therefore, it is of great significance to develop a chlorine-inhibited oxygen evolution electrode, capable of operating in seawater as an electrolyte.

Among recently reported anodic catalysts for seawater electrolysis, manganese dioxide has a combination of relatively high catalytic activity and oxygen evolution selectivity. For instance, Sang et al. reported that doped γ-MnO_2_ has high oxygen evolution selectivity. However, the active material was supported on IrO_2_ containing titanium plate and the catalytic performance was controlled by IrO_2_ and γ-MnO_2_ [16]. While similar electrodes were also reported by Vos et al. [17], high cost and scarcity of IrO_2_ significantly impedes its large-scale application in water electrolysis, requiring the development of novel non-noble metal catalysts. Recently, it was reported that binder-free three-dimensional electrodes exhibit significant electrochemical activity due to their high conductivity [18]. A binder-free MnO_2_-based composite electrode demonstrated good energy storage performance in capacitors [19] and was also used to enhance OER in alkaline media [20]. However, no binder-free manganese dioxide-based materials were studied as highly selective OER electrodes in seawater electrolysis.

In this study, δ-MnO_2_ arrays were hydrothermally synthesized on a highly conductive carbon cloth (MnO_2_/CC) to be used as an anode in seawater electrolysis. Iodometric titration was used here to determine the selectivity of OER. After calculating the current contribution of Cl^−^ oxidation, the selectivity of OER was obtained by subtracting the current of Cl^−^ oxidation from the total oxidation current. To explore the most real OER properties in seawater, real seawater was used as an electrolyte to study its related electrochemical properties, which is different from the synthesized seawater used by most researchers. In addition, the effects of transition metals, such as Fe, Co and Ni. doped MnO_2_ on OER selectivity were also investigated in this work.

## 2. Results and Discussions

The XRD diffraction pattern of MnO_2_/CC shown in Figure 1a contains four diffraction peaks corresponding to (001), (002), (110) and (020) crystal planes of δ-MnO_2_ (PDF # 80-1098), confirming its formation on the CC surface. Figure 1b demonstrates uniform surface coverage with δ-MnO_2_ layer which, according to zoom-in SEM images shown in Figure 1c,d comprises 3D-interconnected nanosheets ca. 30 nm thick.

C, Mn and O elements were detected in the survey XPS spectrum of MnO_2_/CC (Appendix A). The Mn 2p XPS spectra presented in Figure 1e exhibit two peaks at binding energies 643.0 and 654.9 eV, corresponding to spin orbitals of Mn 2p_3/2_ and Mn 2p_1/2_, respectively [21]. The peaks at 641.86 and 653.57 eV are attributed to the Mn(II) on the surface of MnO2/CC, and the peaks at 643.14 and 654.82 eV to the Mn (III). Corresponding binding energies are listed in Appendix A. The O 1s XPS spectrum could be fitted into three peaks, corresponding to interlayer oxygen (Mn-O-Mn), adsorbed OH (Mn-OH) and bound water (-H_2_O) [22]. According to the integral area of these fitted peaks, the proportion of Mn-O-Mn, Mn-OH and H_2_O is 51.50%, 24.57% and 23.93% respectively (Appendix A).

Figure 2a demonstrates OER polarization curves of MnO_2_/CC and CC in real seawater electrolysis, where CC exhibits limited OER catalytic activity, possibly due to metal impurities and some functional groups on its surface. The current response of MnO_2_/CC was higher than that of CC, indicating a positive effect of the MnO_2_ coating. At a current density of 10 mA cm^−2^, the ORE overpotential on the MnO_2_/CC electrode reached 815 mV and its Tafel slope-295 mV/dec, higher than previously reported values obtained in synthetic seawater [23,24,25]. Increased electrode polarization in seawater may be caused by much lower concentration of OH^−^ than that in synthetic analogs, prepared using KOH electrolyte which can accelerate OER reaction kinetics [14]. Additionally, as shown in Appendix A, Ca^2+^ and Mg^2+^ present in seawater produce white precipitates on the electrode, covering much of its active surface sites and reducing electrolysis efficiency. The seawater used in this study contained 0.5 mol L^−1^ of Cl^−^ which participated in the ClER competing with the OER on the anode. As shown in Appendix A, chlorine participates in the following anodic reactions: 2Cl^−^→Cl_2_ + 2e^−^ at pH 0~3, Cl^−^ + H_2_O→HClO + H^+^ + 2e^−^ at pH 3~7.5 and Cl^−^ + 2OH^−^→ClO^−^ + H_2_O + 2e^−^ at pH > 7.5. Released chlorine can also poison the anode, greatly reducing its electrocatalytic performance.

The OER selectivity of X-MnO_2_/CC under real seawater electrolysis conditions was determined by analyzing concentrations of ClO^−^ in the anodic and absorption solutions collected before and after electrolysis (Figure 2c). As shown in Appendix A, plenty of bubbles are released from the electrode surface and no particles dislocated from it, proving good structural stability of the electrode at high current densities. In Appendix A, the gas absorption device containing 1 M NaOH, which can react with chlorine according to the equation 2NaOH + Cl_2_ = NaCl + NaClO + H_2_O, slowly releases the gas delivered from the anode, which improves accuracy of the OER selectivity calculation. The color changes during the titration of the initial anodic and absorption solutions, and those collected after 30 min of electrolysis at 100 mA are shown in Figure 2d–h. Initially, the anodic solution was colorless and transparent (Figure 2d), becoming yellow-brown after the addition of the KI solution (Figure 2e), due to the oxidation to iodine by ClO^−^ (2KI + 2H^+^→2K^+^ + 2HI; 2HI + ClO^−^→I_2_ + Cl^−^ + H_2_O). With the addition of Na_2_S_2_O_3_, the yellow-brown color gradually faded and transformed to light yellow (I_2_ + 2Na_2_S_2_O_3_→2NaCl + Na_2_S_4_O_6_). At that point, exact determination of the titration end point was aided by the addition of the starch solution making the color change more obvious. As shown in Figure 2g, the color turned dark blue, indicating the presence of remaining iodine (nI_2_ + (C_6_H_10_O_5_)6n = (C_18_H_30_O_5_I)_2n_). Na_2_S_2_O_3_ dropwise addition was continued until the solution became colorless and transparent (Figure 2h), at which point the volume of consumed Na_2_S_2_O_3_ for the oxidation of Cl^−^ to ClO^−^ was ascertained. The solution collected in the absorption compartment was similarly titrated until its color was consistent with that of the anodic solution. The selectivity of oxygen evolution over MnO_2_/CC was calculated to be 66.3%, by subtracting the quantity of electricity used for oxidizing Cl^−^ from the total electricity consumed during the 30 min experiment. The MnO_2_/CC electrode maintained its original physical appearance, indicating its strong resistance to chlorine evolution and possible applicability as an anode in seawater electrolysis processes.

In an attempt to improve the OER selectivity of MnO_2_/CC, ions of transition metals such as Fe, Co and Ni were intercalated into MnO_2_/CC using the procedure we developed previously [20]. According to Figure 3a,b, after the metal intercalation, the morphology of Fe-MnO_2_/CC was similar to that of MnO_2_/CC. The crystalline structure of Fe-MnO_2_/CC also has not changed, and its XRD pattern still contained characteristics of δ-MnO_2_ (Figure 3c). The transmission electron microscope (TEM) image (Figure 3d) shows that Fe-MnO_2_ formed on CC is a uniform thin layer structure, implying that more active sites can form on the surface. The lattice fringe with plane spacings of 0.680 and 0.232 nm corresponds to the (001) and (222) planes of MnO_2_ respectively (Figure 3e). Fe atoms embedded in MnO_2_ nanosheets were detected by EDS (Figure 3f) and their even distribution is evident in the elemental mapping image shown in Figure 3j, which, alongside Mn and O mappings of Figure 3h,i, represents the STEM-observed area displayed in Figure 3g. The optimal ion exchange time of 1 h was established by measuring OER selectivity of different Fe-MnO_2_/CC samples, as shown in Appendix A. Using the same synthesis method and 1 h ion exchange time, Co-MnO_2_/CC and Ni-MnO_2_/CC samples were also prepared to study the effect of different transition metal ions on the OER selectivity.

OER polarization curves (Figure 4a) of Fe-MnO_2_/CC, Co-MnO_2_/CC, Ni-MnO_2_/CC and MnO_2_/CC, recorded in seawater, demonstrate that the introduction of transition metal ions in MnO_2_ nanosheets has no positive effect on their electrocatalytic activity towards OER. Overpotentials of Fe-MnO_2_/CC, Co-MnO_2_/CC and Ni-MnO_2_/CC electrodes at 50 and 100 mA cm^−2^, shown in Figure 4b, are slightly higher than that of MnO_2_/CC and corresponding Tafel curve slope values (Figure 4c) are also increased. At 66.3%, the OER selectivity of MnO_2_/CC was the highest among all samples (Appendix A), consistent with polarization curve results. The 3D interconnected nanosheet structure remains unchanged after electrochemical testing (Appendix A). In addition, the Tafel slope measured in real seawater electrolyte is obviously higher than the normal value obtained in 1M KOH (Figure 2b and Figure 4c). It was reported that the relationship between Tafel slope, overpotential and current density can be expressed by the following equation: dlog (j)/d η = 2.303 RT/α nF, showing that Tafel slope is inversely proportional to charge transfer coefficient (α), namely the catalyst with high charge transfer ability has a small Tafel slope. Compared with 1M KOH electrolyte, the adsorption of pollutants, such as microorganisms, bacteria and small particles, in real seawater on the catalyst surface leads to degradation of catalytic performance and hinders the catalytic reaction, resulting in a larger Tafel slope [26].

To explain the negative effect of Fe, Co and Ni on the OER selectivity of MnO_2_/CC in seawater electrolysis, XPS was used to analyze chemical states of X-MnO_2_/CC materials. The survey XPS spectrum (Appendix A) contains characteristic peaks of Fe, Co and Ni. Two peaks at 711.9 and 724.5 eV, corresponding to the Fe 2p_3/2_ and Fe 2p_1/2_ signals of Fe^3+^, with a spin energy interval of 12.6 eV, are visible in the Fe 2p XPS spectrum shown in Appendix A [27]. Similarly, there are two peaks at 781.4 and 796.4 eV in the Co2p spectrum (Appendix A), corresponding to Co 2p_3/2_ and Co 2p_1/2_ signals of Co^2+^ [28]. The Ni 2P XPS spectrum indicates the presence of Ni^2+^ in Ni-MnO_2_/CC [29]. As shown in these high resolution XPS spectra, the valence state of metal cations did not change, indicating that there was no redox reaction in the intercalation process. At the same time, it also confirms that metal ions are successfully embedded in manganese dioxide.

Fitted Mn 2p XPS spectra of Fe-MnO_2_/CC, Co-MnO_2_/CC and Ni-MnO_2_/CC are shown in Figure 5a–c, indicating Mn valences +three and +four [30], whose peak positions are listed in Appendix A. After cation intercalation, no peaks of other Mn species are found, which indicates that MnO_2_ is retained, but the valence state of Mn fluctuates. Since Mn^4+^ prevailed in MnO_2_/CC (Figure 1e), it can be assumed that the introduction of transition metal ions altered the valence state of Mn. According to Figure 5d, Mn^4+^ and Mn^3+^ accounted for 57.84% and 42.16% of the initial MnO_2_, respectively. After the transition metal introduction, the content percentages of Mn^4+^ and Mn^3+^ changed significantly, especially in Co-MnO_2_/CC, where that of Mn^4+^ dropped to 39.46%, and that of Mn^3+^ increased to 60.57%. This can be explained by the gradual reduction of Mn^4+^ to Mn^3+^ to balance the charges of high valence transition metal ions.

The charge state of corresponding O atoms also changes with the transition of the Mn electron structure. According to Appendix A, high resolution O 1S XPS spectra recorded after the metal ions’ introduction into MnO_2_ can be fitted into three peaks, corresponding to interlayer oxygen (Mn-O-Mn), adsorbed OH (Mn-OH) and bound water (-H_2_O). As shown in Appendix A, embedded metal ions alter the content of Mn-OH and other oxygen species, increasing the oxygen vacancy concentration in X-MnO_2_/CC samples.

The rate-determining step in the mechanism of OER catalysis by manganese oxide is the transition from Mn=O to the MnOOH intermediate (Figure 6a) [31,32]. While it was assumed that Mn^3+^ enhances the adsorption of -OH on the MnO_2_ surface and promotes the formation of the Mn=O intermediate, thus advancing the OER, the current study results can be explained using alternative considerations. The solution pH affects chlorine evolution according to the following equations [33,34,35]:S-OH_2_ ^+^ →S-OH + H^+^,(1)
S-OH_ad_ + Cl→HOCl + e^−^,(2)
HOCl + HCl→Cl_2_ + H_2_O,(3)
S-OH→S-O^−^ + H^+^,(4)
where S represents the surface of MnO_2_, reaction two is the rate-determining step and the mechanism of chlorine evolution can be characterized by reaction one. As shown in Figure 6b, Mn adsorbs H_2_O to form the Mn−OH_ad_ intermediate, then adsorbs chloride ions to form Mn−(H)OCl, which subsequently desorbs from the MnO_2_ surface and continues to react with chloride ions in the solution to form Cl_2_. The presence of Mn^3+^ enhances the adsorption of −OH on the manganese oxide surface to form the intermediate Mn−OH_ad_, which reaction Mn−OHad + Cl^−^→Mn−(H)OCl becomes the rate-determining step, greatly accelerating the chlorine evolution process as a result of Fe, Co and Ni introduction in MnO_2_. According to fitted XPS spectra, the content of Mn^3+^ in MnO_2_ increases in the order of MnO_2_/CC < Ni-MnO_2_/CC < Fe-MnO_2_/CC < Co-MnO_2_/CC. For these materials, the kinetics of chlorine evolution gradually increased, resulting in the reduction of the OER selectivity. While the OER selectivity of MnO_2_/CC in real seawater electrolysis needs improvement, it is not likely to be achievable by the introduction of transition metal ions.

## 3. Materials and Methods

### 3.1. Pretreatment of Carbon Cloth

CC piece with dimensions of 4 × 6 cm was placed in a PTFE lined autoclave (instrument model, Company, City, state abbreviation if USA, Country) (TKH−100 mL, Xi’an Taikang Biotechnology Co., Ltd., Xi’an, China) into which 50 mL of 65% nitric acid solution was added. The autoclave was heated at 120 °C for 8 h. Pretreated CC was rinsed with deionized water and dried in an oven at 60 °C.

### 3.2. Preparation of MnO_2_/CC

1 mmol (0.16 g) of KMnO_4_ was dissolved in 50 mL of deionized water and transferred into the PTFE lined autoclave. A 2 × 3 cm piece of pretreated CC was placed in the autoclave, followed by heating at 120 °C for 3 h. The obtained sample was rinsed thoroughly with deionized water and dried in an oven at 60 °C. The sample was labeled as MnO_2_/CC.

### 3.3. Preparation of X-MnO_2_/CC (X = Fe, Co, Ni)

0.8 mmol of the salt containing Fe, Co or Ni was dissolved in 50 mL of deionized water and MnO_2_/CC was immersed in the obtained solution for 1 h. The sample was rinsed in deionized water 3 times and dried in an oven at 60 °C. The obtained material was labelled as X-MnO_2_/CC (X = Fe, Co, Ni).

### 3.4. OER Selectivity Investigation

Constant current at 100 mA cm^−2^ was applied to a three-electrode cell for 30 min, where real seawater was used as an electrolyte, platinum mesh as a counter electrode, and saturated Hg/HgO as a reference electrode. After the initial experiment, 10 mL of the anodic solution was collected and 15 mL of 0.5 M KI was added to it to observe the color change. The discoloration pointed to the presence of hypochlorous acid in the anodic solution, whose concentration was titrated using 0.02 M sodium thiosulfate until light yellow in color and then with 1% starch solution until the color disappeared. The volume of added sodium thiosulfate solution was used to calculate the hypochlorite concentration according to the following equation: n_OCl_^−^ = (V_Thio_·C_Thio_)/V_sample_·V_total_.

n_OCl_^−^: the amount of hypochlorite; V_Thio_: the volume of sodium thiosulfate consumed during the titration; C_Thio_: the molarity of a substance of sodium thiosulfate; V_sample_: the volume of anodic fluid to be titrated; V_total_: the total volume of solution in the anode region.

### 3.5. Physical Characterizations

Scanning electron microscope (SEM, ULTRA plus, Carl Zeiss, Germany) was used to characterize the morphology, and X-ray diffraction (XRD, XD−3A, Shimadzu, Japan) was used to analyze the crystal structure and composition of the prepared materials (angle range: 5–90°, scanning rate: 5° min^−1^). X-ray photoelectron spectroscopy (XPS, VG Escalab210, VG Scientific, Britain) was carried out to analyze the electronic structure and composition of the prepared materials.

### 3.6. Electrochemical Characterizations

The electrochemical performance of the samples was evaluated using a standard three-electrode cell connected to CHI 760E electrochemical workstation (Shanghai Chenhua Instrument Co., Ltd., Shanghai, China). In the three-electrode cell, X-MnO_2_/CC (size: 1 × 1 cm) was used as the working electrode, graphite rod as the counter electrode and Hg/HgO as the reference electrode. For comparison, two different electrolytes, 1M KOH and real seawater, were used. Before the test, the sample was scanned by cyclic voltammetry until the curves could overlap with each other to remove the impurities that might exist on the electrode surface and expose the active sites. The CV sweep speed is 50 mV/s, and the voltage range is −1~0.2 V (vs. Hg/HgO). Linear scanning voltammetry (LSV) of 90% iR compensated OER were measured at a scanning rate of 5 mV/s. The obtained potential is calibrated to the reversible hydrogen electrode (RHE) via the following equation: E_RHE_ = E_Hg/HgO_ + 0.059 × pH + 0.098. The overpotential (η) of OER is calculated by using the following formula: η (V) = E_RHE_ − 1.23 V. The polarization curve was redrawn as the relationship between η and logarithmic current density (J) to calculate the Tafel slopes via the Tafel equation: η = b log j + a, where j is the current density and b is the Tafel slope. To estimate the electrochemical active surface area, the specific capacitance of the electric double layer was measured by cyclic voltammetry measured at different scanning rates and in the range from −0.2 V to −0.1 V (vs. Hg/HgO). The electrochemical impedance spectroscopy (EIS) was obtained at a current density of 10 mA cm^−2^ in a frequency range of 0.01 Hz–100 kHz.

## 4. Conclusions

A series of MnO_2_ nanosheets were successfully synthesized onto the carbon cloth via a facile hydrothermal method as the anodes for seawater electrolysis. Using real seawater as the electrolyte, the selectivity of the MnO_2_/CC electrode for oxygen evolution was 66.3%. By studying the structure of MnO_2_/CC electrodes doping with transitional metallic cations, it was found that Mn^3+^ in MnO_2_ can enhance the adsorption of -OH on the surface of MnO_2_ to form Mn-OH_ad_, thus speeding up the rate-determining step of Mn-OH_ad_ + Cl^−^→ Mn-(H)OCl during the chlorine evolution processes, which has adverse effects on oxygen evolution reaction. Therefore, increasing the content of Mn (+4) in MnO_2_ is an effective way to improve the OER selectivity of MnO_2_ for seawater electrolysis. At the same time, MnO_2_/CC exhibits good tolerance to chlorine evolution reaction, indicating its promising potential as the anode for seawater electrolysis. This work provides a reference for studying the competition between chlorine evolution reaction and oxygen evolution reaction in seawater electrolysis for hydrogen production.

## Figures and Tables

**Figure 1 molecules-28-00854-f001:**
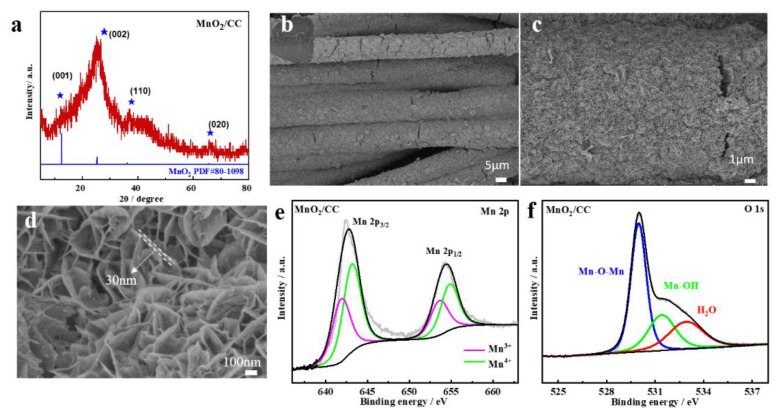
(**a**) XRD pattern of MnO_2_/CC; (**b**–**d**) SEM images of MnO_2_/CC; (**e**) XPS of Mn 2p of MnO_2_/CC; (**f**) O 1s of MnO_2_/CC.

**Figure 2 molecules-28-00854-f002:**
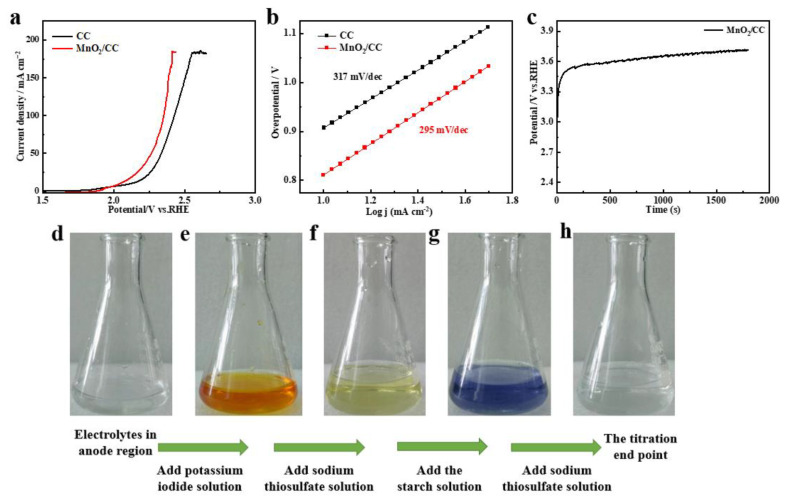
(**a**) OER polarization curves of CC and MnO_2_/CC catalysts in real seawater electrolyte; (**b**) Tafel slope; (**c**) potential vs. time at the current of 100 mA; (**d**–**h**) corresponding color changes during the titration.

**Figure 3 molecules-28-00854-f003:**
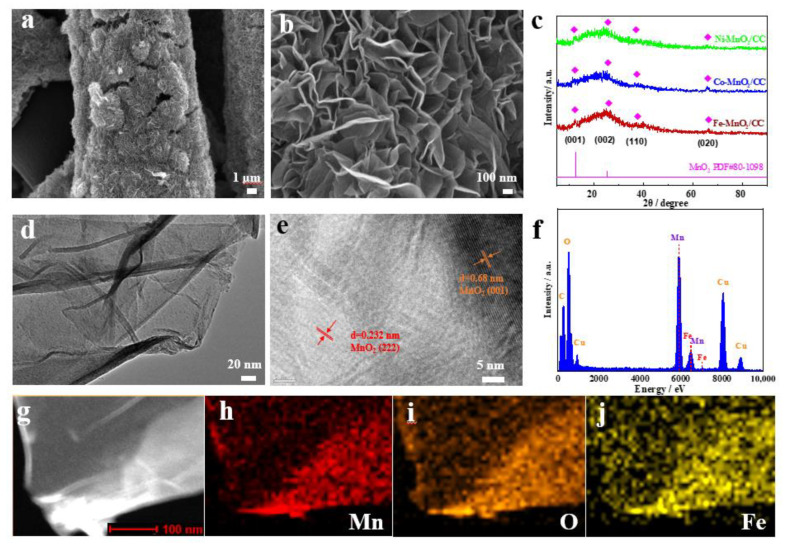
(**a**,**b**) SEM images of Fe-MnO_2_/CC; (**c**) XRD pattern of Fe-MnO_2_/CC; (**d**,**e**) TEM images of Fe-MnO_2_/CC; (**f**) EDS of Fe-MnO_2_/CC; (**g**) STEM image of Fe-MnO_2_/CC; (**h**) Mn elemental mapping of Fe-MnO_2_/CC; (**i**) O elemental mapping of Fe-MnO_2_/CC; (**j**) Fe elemental mapping of Fe-MnO_2_/CC.

**Figure 4 molecules-28-00854-f004:**
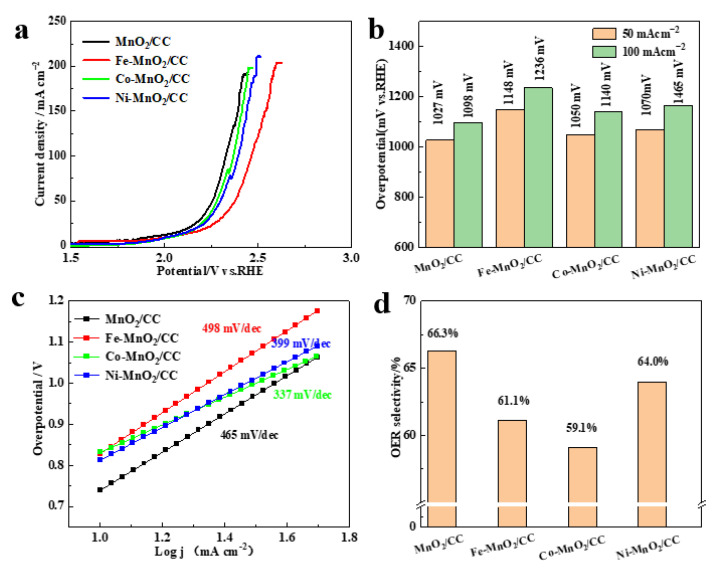
(**a**) OER polarization curves of Fe-MnO_2_/CC, Co-MnO_2_/CC, Ni-MnO_2_/CC and MnO_2_/CC in real seawater electrolyte; (**b**) Overpotentials required for these electrodes to reach current densities of 50 and 100 mA cm^−2^; (**c**) Tafel curves with slope values; (**d**) OER selectivity values.

**Figure 5 molecules-28-00854-f005:**
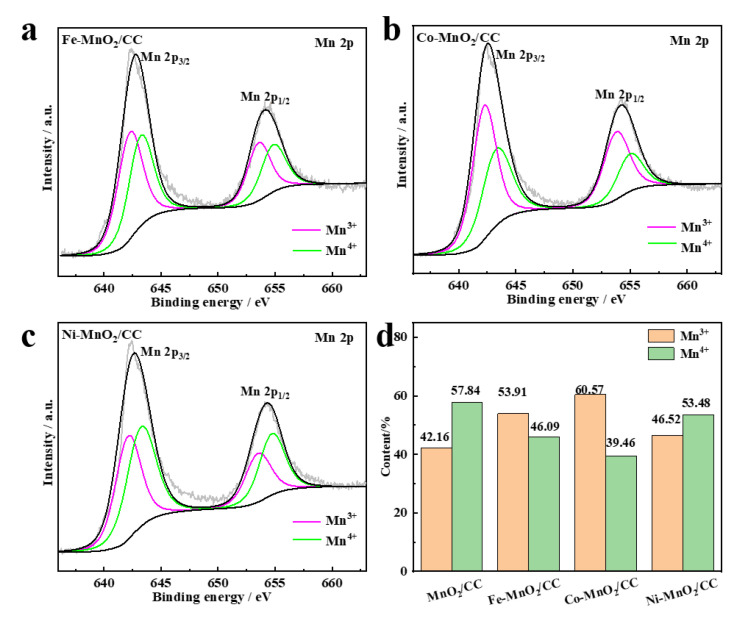
(**a**) High resolution Mn 2p XPS spectra of Fe-MnO_2_/CC; (**b**) High resolution Mn 2p XPS spectra of Co-MnO_2_/CC; (**c**) High resolution Mn 2p XPS spectra of Ni-MnO_2_/CC; (**d**) Histograms of Mn^3+^ and Mn^4+^ in MnO_2_/CC, Fe-MnO_2_/CC, Co-MnO_2_/CC and Ni-MnO_2_/CC.

**Figure 6 molecules-28-00854-f006:**
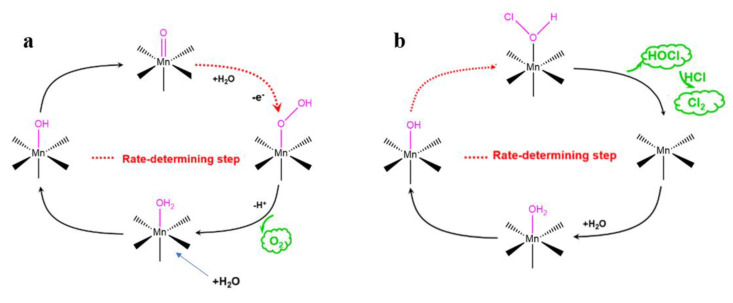
(**a**) Mechanism of OER catalysis by MnO_2_; (**b**) mechanism of ClER catalysis by MnO_2_.

## Data Availability

Not applicable.

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
