# Peer review of "Selectivity of Oxygen Evolution Reaction on Carbon Cloth-Supported δ-MnO2 Nanosheets in Electrolysis of Real Seawater"

_molecules, 2023, doi:10.3390/molecules28020854_

Round 1
Reviewer 1 Report
The manuscript 'Selectivity of oxygen evolution reaction on carbon cloth - supported δ-MnO2 nanosheets in electrolysis of real seawater' reported the preparation of MnO2/CC and its application in sea water electrolysis. The topic of this manuscript is very interesting but the manuscipt could be further improved.
1. The motivation of the paper is week and needs substantial improvement. The novelty of the paper should be emphersized.
2. The authors should explain the XPS results in more details.
3. OER performance in solutions with different Cl concentration may necessarry to investigate the effect of Cl on the cactalytic activity.
Author Response
Response to Reviewer 1 Comments
Dear editors and reviewer:
We are truly grateful to your reviewers’ critical comments and thoughtful suggestions. Those comments are all valuable and helpful revising and improving our paper, as well as important guiding significance to our researches. As such, we have made careful modifications to the original manuscript. Changes to the text regarding specific queries from the reviewers are highlighted in red color. Below you will find our responses to the reviewers’ comments:
Point 1: The motivation of the paper is week and needs substantial improvement. The novelty of the paper should be emphersized.
Response 1: Thank you for your suggestion. We modified the Introduction section based on your comment. All the changes were highlighted in red color.
Point 2: The authors should explain the XPS results in more details.
Response 2: Thank you for your comment. More detail explanation of XPS results were added in the revised manuscript.
Point 3: OER performance in solutions with different Cl concentration may necessarry to investigate the effect of Cl on the cactalytic activity.
Response 3: In this study, we selected seawater with the same Cl concentration for OER test.
Reviewer 2 Report
The work seems to be done well and the authors present interesting and publishable results.
Only comment is that the proof for the existence of delta-MnO2 as the only MnO2 phase present is shallow. I agree that Figure 1a shows clear indications for the presence of delta-MnO2, however the signals are very weak and even somewhat surprising considering the well-developed facets that are visible in Figure 1d.
Moreover, XPS shows a considerable fraction of Mn(3+) whereas in platelet-like MnO2 nanosheets by far the most predominant oxidation state should be 4+. But the deconvoluted spectrum shows considerable amount of Mn(3+) present.
In view of the large number of MnOx polymorphs that can exist and will probably form in a simple uncontrolled hydrothermal process as used here, I think that the authors should weaken the claim that the phase is only delta-MnO2. The fact that xrd shows no other peaks leaves open the possibility of amorphous MnOx phases or minority crystalline MnOx phases, and I think it is realistic to assume so.
A weaker claim would not affect their conclusions on the performance of the MnO2/CC electrode.
Reference 14 is in Chinese characters only and cannot be read by many/most readers. I assume that the paper is in Chinese, but it would help readers to know the author names and title in English at least.
Author Response
Response to Reviewer 2 Comments
Dear editors and reviewer:
We are truly grateful to your reviewers’ critical comments and thoughtful suggestions. Those comments are all valuable and helpful revising and improving our paper, as well as important guiding significance to our researches. As such, we have made careful modifications to the original manuscript. Changes to the text regarding specific queries from the reviewers are highlighted in red color. Below you will find our responses to the reviewers’ comments:
Point 1: Only comment is that the proof for the existence of delta-MnO2 as the only MnO2 phase present is shallow. I agree that Figure 1a shows clear indications for the presence of delta-MnO2, however the signals are very weak and even somewhat surprising considering the well-developed facets that are visible in Figure 1d.
Moreover, XPS shows a considerable fraction of Mn(3+) whereas in platelet-like MnO2 nanosheets by far the most predominant oxidation state should be 4+. But the deconvoluted spectrum shows considerable amount of Mn(3+) present.
In view of the large number of MnOx polymorphs that can exist and will probably form in a simple uncontrolled hydrothermal process as used here, I think that the authors should weaken the claim that the phase is only delta-MnO2. The fact that xrd shows no other peaks leaves open the possibility of amorphous MnOx phases or minority crystalline MnOx phases, and I think it is realistic to assume so.
A weaker claim would not affect their conclusions on the performance of the MnO2/CC electrode.
Reference 14 is in Chinese characters only and cannot be read by many/most readers. I assume that the paper is in Chinese, but it would help readers to know the author names and title in English at least.
Response 1: Thank you for your comments. Reference 14 is in Chinese. According to your suggestion, we have added its English title in the revised manuscript.
Reviewer 3 Report
In this manuscript, the authors reported the selectivity of oxygen evolution reaction on carbon cloth - supported δ-MnO2 nanosheets in electrolysis of real seawater. The findings from this work are interesting. I suggest accepting this work after minor revisions.
1. TEM images should be given to confirm the crystal structure.
2. More information after electrochemical testing should be provided.
3. The selectivity of the newly prepared anode towards the oxygen evolution reaction (OER) was 66.3% after 30 min of electrolyser operation. Is this a good performance?
4. What happened for CC when the potential over 2.5 V (Figure 2a)?
5. The value of Tafel slope is so big. Please provide more explanations.
6. Why δ-MnO2 nanosheets is a good electrocatalysts?
7. Too many mistakes were found. Please carefully check and revised.
8. The following papers (Chem. Soc. Rev., 2022, 51, 4583-4762; Energy & Environmental Materials 2022, e12441 (DOI:10.1002/eem2.12441); Advanced Functional Materials, 2021, 31, 2009779) are recommended to be cited for improving the manuscript.
Author Response
Response to Reviewer 3 Comments
Dear editors and reviewer:
We are truly grateful to your reviewers’ critical comments and thoughtful suggestions. Those comments are all valuable and helpful revising and improving our paper, as well as important guiding significance to our researches. As such, we have made careful modifications to the original manuscript. Changes to the text regarding specific queries from the reviewers are highlighted in red color. Below you will find our responses to the reviewers’ comments:
Point 1: TEM images should be given to confirm the crystal structure.
Response 1: Thank you for your comment. The corresponding part was modified based on your comment.
Point 2: More information after electrochemical testing should be provided.
Response 2: Thank you for your suggestion. More information related to electrochemical testing was provided in the revised manuscript.
Point 3: The selectivity of the newly prepared anode towards the oxygen evolution reaction (OER) was 66.3% after 30 min of electrolyser operation. Is this a good performance?
Response 3: The higher the OER selectivity value, the stronger the oxygen evolution competitiveness, which is a good performance for the electrolysis in the real seawater.
Point 4: What happened for CC when the potential over 2.5 V (Figure 2a)?
Response 4: Thank you for your question. When the voltage exceeds 2.5 V, the polarization curve of CC shows a horizontal trend, which is a normal phenomenon when the working electrode reaches its highest current density.
Point 5: The value of Tafel slope is so big. Please provide more explanations.
Response 5: Thank you for your suggestion. The reason why Tafel's slope is so large was explained in the revised manuscript.
Point 6: Why δ-MnO2 nanosheets is a good electrocatalysts?.
Response 6: Thank you for your question. In such 2D layered structure, δ-MnO2 has a larger specific surface area and can expose more active sites compared with other types of MnO2, which can improve interfacial charge transfer and in turn result in effective catalysis.
Point 7: Too many mistakes were found. Please carefully check and revised.
Response 7: Sorry for this. We double-checked the paperand tried our best to correct these mistakes.
Point 8: The following papers (Chem. Soc. Rev., 2022, 51, 4583-4762; Energy & Environmental Materials 2022, e12441 (DOI:10.1002/eem2.12441); Advanced Functional Materials, 2021, 31, 2009779) are recommended to be cited for improving the manuscript.
Response 8: Thank you for your suggestion. The above latest related papers were cited in the revised manuscript.
Round 2
Reviewer 1 Report
The authors addressed all of my concerns.